# Tolerance and Physiological Responses of Citrus Rootstock Cultivars to Boron Toxicity

**Wanyun Yang [1], Huidong Yang [2,†], Lili Ling [1], Changpin Chun [1] and Liangzhi Peng [1,*]**

[1] Citrus Research Institute, Southwest University—Chinese Academy of Agricultural Sciences, Chongqing 400700, China
[2] Jiangxi Academy of Agricultural Sciences, Nanchang 330000, China
[*] Correspondence: pengliangzhi@cric.cn; Tel.:+86-13500361963
[†] These authors contributed equally to this work.

**Abstract:** Boron (B) is an essential trace nutrient element for citrus, but excessive B levels are frequently encountered in citrus production, potentially resulting in citrus toxicity. To better understand the tolerance and physiological responses of citrus rootstocks to excess B levels, Trifoliate orange, Ziyangxiangcheng, Carrizo citrange, and Red tangerine were treated with four B concentrations (0.05, 0.2, 0.8, and 3.2 mmol/L). High B concentrations resulted in leaf yellowing and shedding and eventual plant death. Chlorophyll content and photosynthetic capacity declined in response to high B concentrations, and relative leaf cell conductivity rose significantly. Trifoliate orange was the first to exhibit symptoms of B toxicity, with the highest levels of B-associated injury. As B concentrations rose, the height increment ratio declined, as did belowground and aboveground dry fresh weight. Soluble protein content initially rose and then fell, while proline content, SOD activity, and POD activity rose with B concentrations. B levels in these rootstocks also increased significantly, with the greatest increases in the leaves. Principal component analysis and subordinate function results revealed that the relative rank order for the B tolerance of citrus rootstocks was: Red tangerine > Carrizo citrange > Ziyangxiangcheng > Trifoliate orange.

**Keywords:** citrus rootstock; B toxicity; physiological response; tolerance




## 1. Introduction

Boron (B) is an essential micronutrient for citrus plants [1], and B deficiency can result in the swelling and cracking of veins together with citrus fruit and capsule deformities [2,3]. To prevent B deficiency, various boric fertilizers, including borax and boric acid, are applied to orchards, but the appropriate B concentration range for citrus plants is relatively narrow [4]. Supplemental borax fertilizer application has been widespread in Chinese citrus orchards over the past several decades [5], and the excessive or irrational application thereof has led to issues associated with excessive B levels in the soil and citrus trees in Jiangxi [6], Guangxi [7], Sichuan [8], Chongqing [9], and Yunnan [10].

Given its status as a micronutrient, B plays key roles in a range of physiological and biochemical processes, including carbohydrate metabolism, carbon dioxide assimilation, photosystem II function, and the maintenance of the antioxidant system [11,12]. When B levels are overly high, these processes can become dysregulated in a manner that is deleterious to plant growth and development. Many prior studies have demonstrated that citrus rootstocks can shape the B uptake, transport, utilization, and tolerance of cultivated plant varieties [13–17]. The tolerance of different rootstocks to B under field conditions varies. In a prior report, we demonstrated clear evidence of B toxicity in 'Orah' mandarins grafted on Trifoliate orange rootstock (*Poncirus trifoliate* (L.) Raf), whereas 'Orah' mandarins grafted on Ziyangxiangcheng rootstock (*Citrus junos* (Sieb.)) under the same production management conditions in the same citrus orchard did not exhibit any symptoms. Rootstock selection in the context of Chinese citrus production is dependent on a range of factors,

including climatic conditions, soil properties, and citrus varieties. Trifoliate orange, Carrizo citrange (*Citrus sinensis* Osb. × *Poncirus trifoliate* Raf), and Red tangerine (Citrus tangerine Hort. ex Tanaka) are rootstocks that are frequently used in the context of citrus production. Ziyangxiangcheng is an emerging hybrid rootstock variety that offers advantages, including rapid growth, early production, and good grafting compatibility. Efficient utilization of rootstocks that can tolerate high B levels may help to mitigate the adverse effects of B toxicity on citrus production. However, most studies conducted to date have primarily focused on analyzing B content in different plant tissues or on issues associated with B deficiency. In contrast, research focused on the relative tolerance of different rootstocks to excess B levels is lacking. Here, rootstocks that are frequently employed for citrus production, including Trifoliate orange, Ziyangxiangcheng, Carrizo citrange, and Red tangerine, were treated with a range of elevated B concentrations. The injury-related symptoms, physiological changes, and biochemical alterations associated with different treatments in these rootstocks were then compared with the goal of characterizing the responses of these rootstocks to excess B exposure, thereby enabling the identification of key determinants of B tolerance in order to provide a valuable reference for rootstock selection efforts for the purposes of citrus production.

## 2. Materials and Methods

### 2.1. Plant Materials and Treatments

This study was conducted at the Citrus Research Institute of Southwest University. The Trifoliate orange, Red tangerine, Ziyangxiangcheng, and Carrizo citrange citrus rootstocks were tested in these analyses. The ripe fruits of four rootstocks were collected from the National Citrus Germplasm Resources Nursery from Southwest University. The seeds were collected, the pectin was removed with quicklime, and they were then washed for later use. Rootstock seeds of similar size were selected, washed, peeled, sterilized, and germinated, followed by transportation into medium consisting of a 1:1 mixture of quartz sand and perlite. Seedlings were cultured in a culture room at 25 °C under photon photosynthetic flux density (PPFD) of 150 $\mu mol \cdot s^{-1} \cdot m^{-2}$, with a 12 h light/12 h dark cycle. When seedlings had reached the 2–3 true leaves stage, they were watered once per week with $\frac{1}{2}$ of the modified Hoagland nutrient solution and with distilled water the rest of the time, with all other culture conditions remaining unchanged. After a 150-day growth period, sixty neat seedlings of similar size were selected for each rootstock and transplanted into a perforated plastic incubator (59 cm × 45.5 cm × 32 cm), with culture medium consisting of a 1:1 mixture of clean river sand and quartz sand. Boxes were placed into a colorless, transparent, rainproof, ventilated plastic shed. The B solution used for treatment was analytical-pure borax ($Na_2B_4O_7 \cdot 10H_2O$, Sinopharm Chemical Reagent Co., Ltd., Shanghai, China). Seedlings were treated with B at concentrations of 0.05 mmol/L (CK), 0.2 mmol/L, 0.8 mmol/L, and 3.2 mmol/L (B 0.05, B 0.2, B 0.8, and B 3.2, respectively). Treatments were performed in 3 replicates, with 10 seedlings per replicate. Every 2 weeks, each pot was treated with 4 L of modified Hoagland nutrient solution, and 2 L of the appropriate B treatment solution was added every 3 days. Medium was rinsed on a monthly basis with distilled water to protect against excessive mineral nutrient accumulation. When over 50% of plants had died under the highest tested B levels (B 3.2; 52 DAT (days after treatment)), treatment was completed and all plants were harvested.

### 2.2. B Injury Symptom Assessment and B Injury Index Calculation

Rootstock leaf wilting was assessed on a daily basis and the date, grade, and other wilting-related symptoms were recorded. Plant-growth indices were measured before and following treatment [18]. After treatment, plant height was measured, and the aboveground and belowground plant portions were separated, with the fresh weight (FW) being measured following washing and drying. Enzymatic activity was disrupted by heating samples for 30 min at 105 °C, followed by drying to constant weight at 75 °C. Sample dry

weight (DW) was then measured. The height increment ratio was calculated based on the plant height before and after treatment as follows:

$$\text{Height increment ratio} = \frac{\text{measured value after treating} - \text{measured value before treating}}{\text{measured value before treating}}$$

The B toxicity level standard was determined based on the citrus salt injury classification standard [19,20] and Lidia Aparicio-Durán's citrus B injury index [21]. Citrus rootstock B injury was graded as level 0 (normal growth), level 1 (mild damage, with limited yellowing, scorching, and mottling of leaf tips and margins), level 2 (moderate damage, with large amounts of yellow or brown spots on leaves, half of leaf tips being scorched or yellowed, or a small number of leaves falling off or wilding), level 3 (severe damage, with most leaves appearing scorched and about $\frac{1}{2}$ of the leaves falling off), and level 4 (very severe damage, with all of the leaves falling off branches dying, and potentially plant death).

$$\text{B injury index} = \frac{\sum(\text{injury grade} \times \text{number of plants in a level})}{\text{total number of plants investigated} \times \text{highest injury level}} \times 100\%$$

### 2.3. Measurements of Physiological and Biochemical Parameters

Absolute leaf pigment content was measured on 30 DAT [22], while SPAD values were measured at 8, 21, 35, and 49 DAT with a handheld SPAD-502 chlorophyll meter (KONICA MINOLTA, Japan). On 8, 21, and 35 DAT, leaf relative conductivity was measured as per methods reported by Liu et al. [23]. On 8, 24, 32, and 45 DAT, photosynthetic indices for mature leaves under clear weather conditions were measured from 9:00 to 11:00 with a Li-6400 portable photosynthesis instrument (Li-COR company, Lincoln, NE, USA). When leaves of plants under the B 3.2 treatment conditions exhibited clear evidence of toxicity but were not yet dead (30–35 DAT), samples were collected. The superoxide dismutase (SOD) levels in these leaves were measured via the nitroblue tetrazolium photoreduction method, while catalase (CAT) activity was measured via the guaiacol method, and peroxidase (POD) activity was measured via the hydrogen peroxide decomposition method [12]. Soluble protein content was determined through the Bradford method, using BSA as standard, and proline content was measured via the acid ninhydrin colorimetric method [13].

### 2.4. Measurement of B Levels in Plants

At the end of the treatment period, the leaves, stems, and roots of plants from the B 0.05 and B 3.2 conditions were collected. Samples were then pretreated and B concentrations therein were measured via the Ashing-formamide colorimetric method, as reported previously by Ling et al. and LY/T 1273-1999 [6,24].

### 2.5. Data Processing and Formula Calculations

Data analyses and principal component analyses were performed using SPSS 19.0. Results were compared with ANOVAs and Duncan's multiple range test. GraphPad Prism 9.2 was used for figure construction. The B tolerance of these four citrus rootstock seedlings was evaluated with a subordinate function, as reported previously [25–27]. Based on the measured indices, the average values for the B treatment group and the control group were calculated, and the B resistance coefficients for each index were obtained, after which correlation analyses were performed to obtain a correlation coefficient matrix. Principal component analysis was conducted for the B resistance coefficients, and the original index was converted into a new independent comprehensive index.

$$\text{B resistance coefficient} = \frac{\text{treatment measured value}}{\text{control measured value}} \times 100\% \qquad (1)$$

Formula (2) was used to calculate the D value of B tolerance for these four citrus rootstocks under excessive B stress conditions obtained through the comprehensive evaluation index. A large D value corresponded to greater rootstock B tolerance.

$$D = \sum_{j=1}^{n} \left[ U(X_j) * W_j \right] j = 1, 2, \ldots\ldots, n \tag{2}$$

In Formula (2)

$$U(Xj) = (Xj - X_{\min})/(X_{\max} - X_{\min}) j = 1, 2, \ldots\ldots, n \tag{3}$$

$$W_j = P_j / \sum_{j=1}^{n} P_j \, j = 1, 2, \ldots\ldots, n \tag{4}$$

The subordinate function value for each comprehensive index was calculated with Formula (3), where $X_j$ is the $j$th comprehensive index, $X_{\min}$ is the smallest value of the $j$th index, and $X_{\max}$ is the largest value of the $j$th index. Formula (4) was used to calculate the weight of each comprehensive index based on the comprehensive index contribution rate. The $W_j$ value represents the importance of the $j$th comprehensive index, while $P_j$ is the contribution rate of the $j$th comprehensive index of each rootstock.

## 3. Results

### 3.1. B Injury Symptoms and Injury Indices in Citrus Rootstocks Exposed to Excess B Stress

When exposed to excess B concentrations, citrus rootstocks exhibited yellowing of the leaf tips and macular chlorosis along the leaf margins during the early stages of toxicity. Over time, the macula gradually expanded as the severity of B toxicity increased until the main veins had turned yellow, the leaves fell off, the top branches died, and the entire plant eventually died (Figure 1). Symptoms of injury in these citrus rootstocks can serve as a measure of B tolerance, with the time to the initial occurrence of these symptoms being negatively correlated with B tolerance.

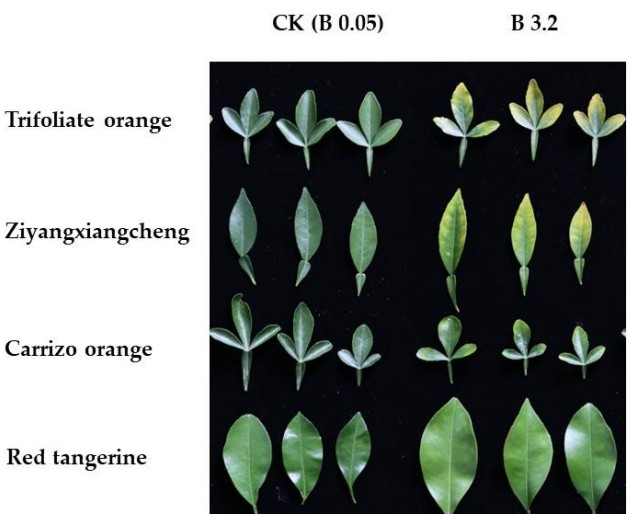

**Figure 1.** The toxicity symptoms of leaves under B 3.2 for four citrus rootstocks in 42 DAT.

The earliest symptoms of injury were observed in Trifoliate orange at 19 DAT at the highest B concentration of 3.2 mmol/L (B 3.2), at 28 DAT for B 0.8, and at 52 DAT for B 0.2 (Table 1). Under B 3.2 treatment conditions, Ziyangxiangcheng and Carrizo orange seedlings, respectively, exhibited symptoms at 21 and 24 DAT, with corresponding symptom onset under B 0.8 treatment conditions at 30 DAT and 39 DAT. Neither of these rootstock seedlings developed symptoms under B 0.2 treatment conditions at any point

during the study period. Symptom onset in Red tangerine occurred at the latest time point. No level 4 injury symptoms were observed in Red tangerine seedlings exposed to B 3.2 treatment conditions, and no plants died over the study period. Under B 0.8 conditions, these seedlings only exhibited level 1 and 3 injury symptoms, while no symptoms of B toxicity were evident under the B 0.2 treatment conditions.

**Table 1.** The time to the occurrence of symptoms of B injury in four citrus rootstock seedlings.

| B Concentration (mmol/L) | Rootstocks | The Time to the Occurrence of B Injury Symptoms/DAT | | | |
|---|---|---|---|---|---|
| | | Level 1 | Level 2 | Level 3 | Level 4 |
| 0.05 | Trifoliate orange | — | — | — | — |
| | Ziyangxiangcheng | — | — | — | — |
| | Carrizo citrange | — | — | — | — |
| | Red Tangerine | — | — | — | — |
| 0.2 | Trifoliate orange | 52 | — | — | — |
| | Ziyangxiangcheng | — | — | — | — |
| | Carrizo citrange | — | — | — | — |
| | Red Tangerine | — | — | — | — |
| 0.8 | Trifoliate orange | 28 | 47 | — | — |
| | Ziyangxiangcheng | 30 | 50 | — | — |
| | Carrizo citrange | 39 | — | — | — |
| | Red Tangerine | 50 | — | — | — |
| 3.2 | Trifoliate orange | 19 | 24 | 28 | 39 |
| | Ziyangxiangcheng | 21 | 30 | 42 | 42 |
| | Carrizo citrange | 24 | 32 | 47 | 50 |
| | Red tangerine | 28 | 42 | 52 | — |

Over the treatment period, all rootstocks grew readily under control (B 0.05) conditions, and Trifoliate oranges were the only rootstock to exhibit mild symptoms of B toxicity under the B 0.2 treatment conditions (Table 1). As such, B injury index values for these rootstocks were only calculated for the B 0.8 and B 3.2 treatment conditions (Table 2). These analyses revealed significant differences in B injury indices among these different citrus rootstocks. Under B 3.2 conditions, Trifoliate orange exhibited the highest injury index value (91.25%), while respective values for Carrizo citrange and Red tangerine were just 61.25% and 48.75%. When B concentrations were increased from 0.8 to 3.2 mmol/L, the B injury index values for Trifoliate orange and Red tangerine at 52 DAT rose by 53.75% and 42.5%, respectively.

**Table 2.** B injury index values for four citrus rootstock seedlings grown under excess B stress conditions.

| B Concentration (mmol/L) | Rootstocks | The B Injury Index/% | | |
|---|---|---|---|---|
| | | 43 DAT | 52 DAT | Mean |
| 0.8 | Trifoliate orange | 25 | 50 | 37.5 |
| | Ziyangxiangcheng | 22.5 | 32.5 | 27.5 |
| | Carrizo citrange | 5 | 20 | 12.5 |
| | Red tangerine | 0 | 12.5 | 6.25 |
| 3.2 | Trifoliate orange | 82.5 | 100 | 91.25 |
| | Ziyangxiangcheng | 80 | 92.5 | 86.25 |
| | Carrizo citrange | 32.5 | 90 | 61.25 |
| | Red tangerine | 30 | 67.5 | 48.75 |

These results, thus, indicated that Trifoliate orange is the rootstock with the highest level of B sensitivity, followed by Ziyangxiangcheng and Carrizo citrange, with Red tangerine exhibiting the highest levels of B tolerance.

### 3.2. The Impact of Excess B on Rootstock Growth

As the levels of excess B stress rose, the DW and FW values for both the aboveground and belowground portions of these rootstocks all decreased (Table 3). Under B 0.2 and B 0.8 conditions, slight decreases in the DW and FW of the aboveground and belowground parts of these seedlings were observed relative to appropriate controls. Under B 3.2 conditions, the DW and FW of the aboveground and belowground portions of all of these rootstocks declined significantly, relative to appropriate controls, with the DW of the aboveground portions of Trifoliate orange, Ziyangxiangcheng, Carrizo citrange, and Red tangerine seedlings having decreased by 54.86%, 46.19%, 43.86%, and 36.40%, respectively. In contrast, the root–shoot ratios for all of these rootstocks increased relative to appropriate controls under B 3.2 treatment conditions, while those of the Trifoliate orange, Ziyangxiangcheng, and Carrizo citrange seedlings differed significantly, relative to corresponding controls. This suggested that excess B stress had a greater impact on the aboveground growth of these rootstocks relative to their belowground growth. The height increment ratio for these rootstocks declined significantly with increasing B stress, with the most pronounced reduction under B 3.2 conditions.

**Table 3.** The impact of excess B stress on the fresh weight, dry weight, height increment ratio, and root–shoot ratio of four citrus rootstock seedlings.

| Rootstocks | B Concentration (mmol/L) | Aboveground FW/g | Underground FW/g | Aboveground DW/g | Underground DW/g | Height Increment Ratio/% | Root–Shoot Ratio |
|---|---|---|---|---|---|---|---|
| Trifoliate orange | 0.05 | 9.53 ± 1.06a | 4.68 ± 0.71a | 4.23 ± 0.46a | 1.77 ± 0.22a | 68.84 ± 11.6a | 0.4 ± 0.03b |
|  | 0.2 | 9.31 ± 0.73a | 3.81 ± 0.06a | 4.29 ± 0.43a | 1.4 ± 0.1a | 67.62 ± 5.32a | 0.33 ± 0.01c |
|  | 0.8 | 9.21 ± 1.97a | 3.76 ± 1.45a | 4.10 ± 0.85a | 1.4 ± 0.34a | 61.68 ± 18.27a | 0.34 ± 0.05b |
|  | 3.2 | 4.05 ± 1.11b | 3.24 ± 1.11a | 1.91 ± 0.56b | 1.05 ± 0.34a | 40.58 ± 5.42b | 0.55 ± 0.07a |
| Ziyangxiangcheng | 0.05 | 25.27 ± 2.92a | 11.31 ± 1.13a | 10.19 ± 1.38a | 3.96 ± 0.46a | 33.88 ± 3.38a | 0.38 ± 0.01b |
|  | 0.2 | 25.84 ± 1.08a | 10.61 ± 0.8a | 10.09 ± 1.12a | 3.67 ± 0.75a | 43.01 ± 5.61b | 0.39 ± 0.04b |
|  | 0.8 | 21.84 ± 1.86ab | 9.82 ± 4.54a | 8.97 ± 4.55a | 3.54 ± 1.67a | 32.74 ± 4.81b | 0.39 ± 0.08b |
|  | 3.2 | 12.71 ± 3.83b | 9.59 ± 3.87a | 5.48 ± 2.84a | 3.22 ± 1.17a | 6.75 ± 5.66c | 0.59 ± 0.15a |
| Carrizo citrange | 0.05 | 11.05 ± 0.80a | 4.14 ± 1.04a | 4.43 ± 0.42a | 1.62 ± 0.18a | 52.34 ± 12.59a | 0.37 ± 0.08b |
|  | 0.2 | 8.96 ± 0.69ab | 3.61 ± 1.00ab | 3.58 ± 0.93ab | 1.48 ± 0.35ab | 65.03 ± 12.06b | 0.41 ± 0.01b |
|  | 0.8 | 7.72 ± 1.8bc | 3.08 ± 0.62b | 3.11 ± 0.77b | 1.18 ± 0.19b | 34.06 ± 4.49c | 0.43 ± 0.07b |
|  | 3.2 | 5.79 ± 0.89c | 3.03 ± 0.33b | 2.49 ± 0.59b | 1.16 ± 0.16b | 17.42 ± 3d | 0.47 ± 0.13a |
| Red tangerine | 0.05 | 9.95 ± 0.86a | 4.45 ± 0.74a | 4.05 ± 0.5a | 1.72 ± 0.23a | 47.49 ± 16.98a | 0.42 ± 0.02a |
|  | 0.2 | 9.84 ± 0.74a | 3.11 ± 0.38ab | 3.93 ± 0.25ab | 1.30 ± 0.64b | 44.69 ± 8.65a | 0.33 ± 0.21a |
|  | 0.8 | 9.50 ± 1.69a | 2.93 ± 0.87bc | 3.48 ± 0.58bc | 1.26 ± 0.21b | 44.44 ± 4.95a | 0.36 ± 0.04a |
|  | 3.2 | 6.47 ± 1.62b | 2.79 ± 0.82c | 2.58 ± 0.64c | 1.12 ± 0.28b | 23.47 ± 14.09b | 0.41 ± 0.06a |

Note: B 0.05 served as the control (CK) concentration, and data were compared with Duncan's multiple range test. Different letters indicate significant differences among B concentrations for the indicated rootstock ($p < 0.05$).

### 3.3. The Impact of Excess B Stress on Rootstock Leaf Photosynthetic Indices

Under control (B 0.05) conditions, leaf SPAD gradually rose for each rootstock, with the exception of a decrease for Carrizo citrange by 3.93 from 35 DAT to 49 DAT under B 0.2 conditions (Figure 2). Under B 0.8 conditions, the SPAD from 0 to 35 DAT did not differ significantly, relative to control conditions. At 49 DAT, the SPAD ratios for Trifoliate orange, Ziyangxiangcheng, Carrizo citrange, and Red tangerine decreased by 16.25, 5.4, 1.89, and 1.26, respectively. At 49 DAT, the SPAD of Trifoliate orange differed significantly from that of control seedlings, but no other differences were significant. From 35 to 49 DAT under B 3.2 conditions, the SPAD of these rootstocks declined significantly, with respective reduction rates in the Ziyangxiangcheng, Carrizo citrange, and Red tangerine seedlings of 50%, 37.39%, and 22.94%, while Trifolaite orange leaves turned yellow and fell off at the point. Similar changes in absolute chlorophyll content were observed in the leaves of these plants (Figure 3). Under B 0.05 conditions, all rootstock leaves maintained a high net photosynthetic rate (Pn). As B concentrations rose, the Pn for each rootstock declined, with the degree of this reduction in Pn being more pronounced as treatment time increased. During this treatment process, Pn declined most substantially in Trifoliate orange seedlings, whereas the drop in Red tangerine seedlings was smallest (Table 4).

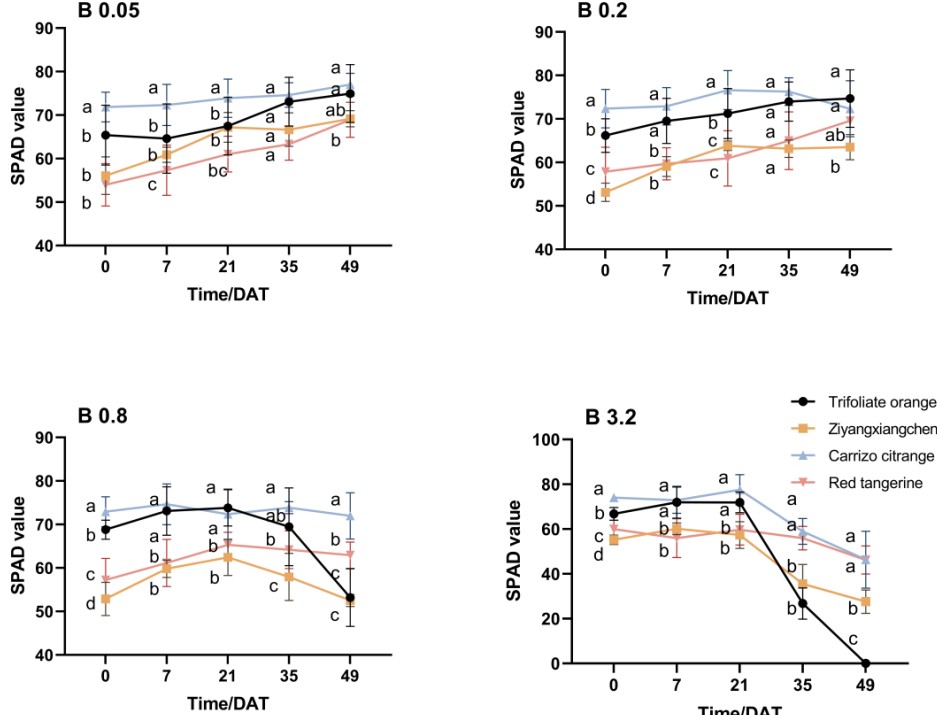

**Figure 2.** The impact of excess B stress on relative chlorophyll content in the leaves of four citrus rootstock seedlings. Note: B 0.05 served as the control (CK) concentration, and data were compared with Duncan's multiple range test. Different letters indicate significant differences between rootstocks at the same DAT ($p < 0.05$).

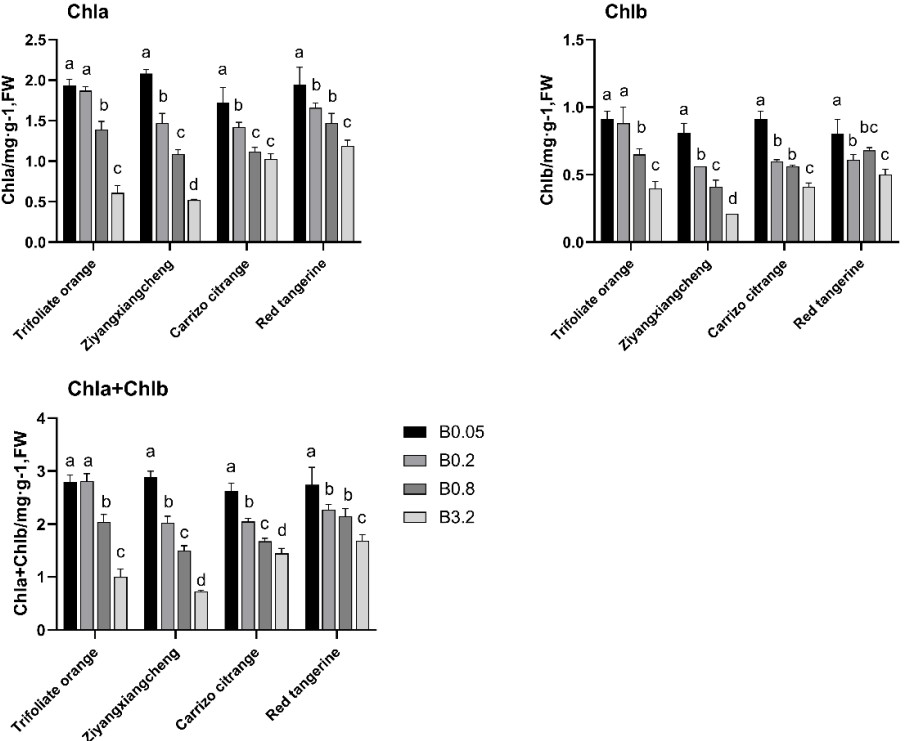

**Figure 3.** The impact of excess B stress on photosynthetic pigment levels in the leaves of four citrus rootstock seedlings. Note: B 0.05 served as the control (CK) concentration, and data were compared with Duncan's multiple range test. Different letters indicate significant differences among B concentrations for the indicated rootstock ($p < 0.05$).

**Table 4.** The impact of excess B stress on the net photosynthetic rates in the leaves of four citrus rootstock seedlings.

| Rootstocks | B Concentration (mmol/L) | Net Photosynthetic Rate/$\mu$mol·m$^{-2}$·s$^{-1}$ | | | |
|---|---|---|---|---|---|
| | | 8 DAT | 24 DAT | 32 DAT | 45 DAT |
| Trifoliate orange | 0.05 | 6.8 ± 1.40a | 6.53 ± 0.70a | 6.73 ± 0.94a | 5.77 ± 1.67a |
| | 0.2 | 6.86 ± 0.76a | 6.81 ± 0.84a | 5.92 ± 1.37a | 5.23 ± 2.11a |
| | 0.8 | 5.80 ± 0.75ab | 5.41 ± 0.09a | 5.27 ± 0.30a | 2.98 ± 0.66b |
| | 3.2 | 3.91 ± 0.15b | 2.76 ± 0.05b | 1.34 ± 0.33b | 0.07 ± 0.06c |
| Ziyangxiangcheng | 0.05 | 6.53 ± 1.42a | 6.08 ± 0.42a | 5.65 ± 0.03a | 4.84 ± 0.55a |
| | 0.2 | 6.08 ± 0.62a | 6.13 ± 0.77a | 6.13 ± 1.31a | 5.61 ± 1.07a |
| | 0.8 | 6.29 ± 0.18a | 3.69 ± 0.00b | 5.46 ± 0.81a | 4.88 ± 1.15a |
| | 3.2 | 3.14 ± 0.28b | 3.28 ± 0.30b | 2.29 ± 0.35b | 0.54 ± 0.25b |
| Carrizo citrange | 0.05 | 6.82 ± 0.48a | 6.79 ± 0.79a | 6.95 ± 0.62a | 5.47 ± 1.54a |
| | 0.2 | 4.54 ± 0.82b | 5.77 ± 0.57b | 5.96 ± 0.80a | 5.68 ± 1.56a |
| | 0.8 | 4.47 ± 0.04b | 4.76 ± 0.65b | 4.47 ± 0.53ab | 4.46 ± 0.3ab |
| | 3.2 | 3.39 ± 0.08b | 3.37 ± 0.43b | 2.46 ± 0.02b | 2.38 ± 0.00b |
| Red tangerine | 0.05 | 6.11 ± 0.17a | 5.91 ± 0.42a | 5.86 ± 0.42a | 6.75 ± 1.44a |
| | 0.2 | 5.06 ± 0.87ab | 5.66 ± 0.31a | 5.16 ± 0.89a | 5.68 ± 3.45a |
| | 0.8 | 4.61 ± 0.01ab | 6.03 ± 0.95a | 5.06 ± 0.24a | 5.46 ± 0.78a |
| | 3.2 | 4.52 ± 0.21b | 4.75 ± 0.10a | 3.22 ± 0.87b | 1.56 ± 0.34b |

Note: B 0.05 served as the control (CK) concentration, and data were compared with Duncan's multiple range test. Different letters indicate significant differences among B concentrations for the indicated rootstock ($p < 0.05$).

### 3.4. The Impact of Excess B Stress on Rootstock Leaf Cell Membrane Permeability

While there was some level of variability in the relative electrical conductivity of the leaves of these four rootstocks under control (B 0.05) conditions, these values generally remained between 8% and 10% (Table 5). At 8 DAT, relative electrical conductivity for each rootstock did not differ significantly from the corresponding control under any treatment conditions. At 21 DAT, the B 0.2 and B 0.8 treatment conditions were not associated with any change in relative electrical conductivity in leaves for any of the tested rootstocks. However, the relative electrical conductivity of leaves from Trifoliate orange, Ziyangxiangcheng, Carrizo citrange, and Red tangerine under B 3.2 conditions at this time point was significantly increased relative to control conditions, with respective 1.82-, 1.46-, 0.81-, and 0.019-fold increases. At 35 DAT, the electrical conductivity of the leaves of each rootstock significantly increased with rising B concentrations, with the highest levels of relative electrical conductivity being evident in Trifoliate oranges, while this increase was smallest in Red tangerine seedlings. However, no significant differences in relative electrical conductivity were observed among these rootstocks.

**Table 5.** The impact of excess B stress on relative electrical conductivity in the leaves of four citrus rootstock seedlings.

| Rootstocks | B Concentration (mmol/L) | Relative Electrical Conductivity/% | | |
|---|---|---|---|---|
| | | 8 DAT | 21 DAT | 35 DAT |
| Trifoliate orange | 0.05 | 8.10 ± 0.19b | 8.82 ± 1.1b | 9.79 ± 0.16b |
| | 0.2 | 8.21 ± 0.19b | 8.20 ± 0.77b | 15.93 ± 0.76b |
| | 0.8 | 9.11 ± 0.02a | 10.31 ± 0.63b | 18.13 ± 0.66b |
| | 3.2 | 9.11 ± 0.11a | 24.89 ± 1.77a | 46.99 ± 9.80a |
| Ziyangxiangcheng | 0.05 | 10.49 ± 0.59ab | 10.49 ± 1.34b | 10.78 ± 0.97b |
| | 0.2 | 8.25 ± 0.25b | 9.60 ± 0.69b | 13.66 ± 0.51b |
| | 0.8 | 10.80 ± 0.36a | 11.01 ± 1.04b | 18.44 ± 2.20b |
| | 3.2 | 8.90 ± 0.63ab | 25.8 ± 2.09a | 37.98 ± 10.9a |
| Carrizo citrange | 0.05 | 7.67 ± 0.32b | 7.18 ± 0.36b | 10.22 ± 1.01b |
| | 0.2 | 9.52 ± 0.14a | 9.63 ± 1.79b | 8.43 ± 0.54c |
| | 0.8 | 8.32 ± 1.31b | 8.58 ± 1.12b | 10.47 ± 0.50b |
| | 3.2 | 8.79 ± 0.26ab | 13.02 ± 1.74a | 18.35 ± 0.70a |
| Red tangerine | 0.05 | 9.58 ± 0.77ab | 10.26 ± 0.47a | 8.18 ± 0.83b |
| | 0.2 | 9.27 ± 0.31ab | 9.78 ± 0.50a | 11.68 ± 2.15a |
| | 0.8 | 8.76 ± 0.19b | 8.06 ± 0.48b | 12.84 ± 0.07a |
| | 3.2 | 10.08 ± 0.57a | 10.46 ± 0.52a | 13.30 ± 0.92a |

Note: B 0.05 served as the control (CK) concentration, and data were compared with Duncan's multiple range test. Different letters indicate significant differences among B concentrations for the indicated rootstock ($p < 0.05$).

### 3.5. The Impact of Excess B Stress on Antioxidant Enzyme Activity and Osmotic Regulators in Rootstock Leaves

To further examine the tolerance characteristics of these different citrus rootstocks, the levels of SOD, CAT, and POD enzymatic activity in the leaves of citrus rootstock seedlings treated with varying B concentrations were next analyzed (Table 6).

**Table 6.** The impact of excess B stress on antioxidant enzyme activity and osmotic regulators in the leaves of four citrus rootstock seedlings.

| Rootstocks | B Concentration (mmol/L) | SOD Activity/$\mu g \cdot g^{-1}$ | POD Activity/$\mu \cdot g^{-1} \cdot min$ | CAT Activity/$\mu \cdot g^{-1} \cdot min$ | Proline Content/$\mu g \cdot g^{-1}$ | Soluble Protein Content/$\mu g \cdot g^{-1}$ |
|---|---|---|---|---|---|---|
| Trifoliate orange | 0.05 | 213.65 ± 8.19b | 27666.12 ± 695.71d | 153.97 ± 20.76a | 121.94 ± 18.92c | 57.27 ± 3.61b |
| | 0.2 | 281.43 ± 3.25b | 33683.05 ± 739.41c | 109.54 ± 1.02b | 133.61 ± 3.79c | 89.82 ± 1.59a |
| | 0.8 | 329.95 ± 17.78a | 60330.02 ± 1092.46b | 59.38 ± 5.16c | 215.01 ± 2.30b | 51.56 ± 5.67b |
| | 3.2 | 445.24 ± 34.17a | 87285.44 ± 3164.95a | 55.86 ± 5.83c | 422.78 ± 6.25a | 30.47 ± 1.35c |
| Ziyangxiangcheng | 0.05 | 540.96 ± 5.43c | 36722.88 ± 861.46a | 228.15 ± 1.60a | 130.50 ± 1.05c | 75.49 ± 3.14b |
| | 0.2 | 635.92 ± 1.92b | 56475.96 ± 975.96b | 195.16 ± 4.04b | 185.74 ± 9.85b | 82.86 ± 1.50a |
| | 0.8 | 667.93 ± 37.60ab | 59082.34 ± 3504.51b | 137.02 ± 6.23c | 190.61 ± 8.29b | 65.19 ± 3.89c |
| | 3.2 | 690.51 ± 4.84ba | 74144.10 ± 2650.28c | 64.75 ± 1.26d | 401.21 ± 4.13a | 36.87 ± 1.08d |
| Carrizo citrange | 0.05 | 429.73 ± 12.72b | 30151.85 ± 309.03c | 237.93 ± 9.34a | 145.59 ± 13.08c | 67.13 ± 2.58b |
| | 0.2 | 456.15 ± 10.00c | 31303.11 ± 440.88bc | 208.12 ± 1.40b | 165.49 ± 1.56b | 71.65 ± 0.36a |
| | 0.8 | 543.24 ± 22.45b | 34293.19 ± 172.82b | 130.08 ± 5.97c | 174.19 ± 3.44b | 66.44 ± 3.21b |
| | 3.2 | 697.77 ± 38.03a | 63850.68 ± 1322.63a | 76.83 ± 2.67d | 269.8 ± 7.16a | 42.82 ± 0.96c |
| Red tangerine | 0.05 | 312.02 ± 26.10a | 22949.96 ± 671.73b | 147.60 ± 3.92a | 77.81 ± 2.57b | 52.2 ± 0.91b |
| | 0.2 | 503.38 ± 11.12b | 25474.78 ± 2704.18b | 111.15 ± 2.12b | 87.09 ± 3.24b | 64.02 ± 3.87a |
| | 0.8 | 512.40 ± 24.53b | 27339.98 ± 919.50b | 108.93 ± 15.21b | 96.33 ± 0.62b | 56 ± 6.33b |
| | 3.2 | 748.47 ± 6.09c | 39282.82 ± 1804.57a | 77.53 ± 1.08c | 164.45 ± 14.87a | 48.76 ± 0.23b |

Note: B 0.05 served as the control (CK) concentration, and data were compared with Duncan's multiple range test. Different letters indicate significant differences among B concentrations for the indicated rootstock ($p < 0.05$).

As B concentrations rose, SOD and POD enzymatic activity levels increased significantly in the leaves of these four rootstock seedlings, whereas CAT activity fell significantly. Under B 3.2 treatment conditions, maximal increases in SOD activity were observed in Red tangerine, whereas the lowest levels were evident in Trifoliate orange seedlings. However, POD activity increased the most in trifoliate orange and was least pronounced in Red tangerine. Consistently, the largest decrease in CAT activity was evident in Trifoliate orange, whereas this decrease was smallest in Red tangerine.

Significant increases in proline levels were observed in the leaves of these rootstock seedlings under conditions of increased B treatment relative to the control (B 0.05) conditions. The highest proline content in Trifoliate orange leaves was observed under the B 0.8 and B 3.2 treatment conditions, with these values having, respectively, increased by 76.32% and 246.71% relative to control conditions. In contrast, these increases were the smallest in Carrizo citrange and Red tangerine seedlings.

Leaf soluble protein content for all of these rootstocks initially rose and then fell with increasing B concentrations, with all treatments reaching maximum levels under B 0.2 treatment conditions before decreasing substantially in response to higher B concentrations (Table 6).

### 3.6. The Impact of Excess B Stress on the B Content in Different Parts of Four Citrus Rootstock Seedlings

Wang N N et al. reported that the distribution of B content in different varieties and tissues of citrus was 10–150 mg/kg [17]. To further reveal the effect of excess B on its distribution, we focused on the content of B in various tissue parts in the control and B 3.2 treatment at 52 DAT. Under control (B 0.05) conditions, B levels in the leaves, stems, and roots of all four rootstocks remained low, with the highest levels in leaves, followed by the stems and roots (Figure 4). Under B 3.2 treatment conditions, the B levels in all parts of these seedlings rose, with 4.99-, 5.96-, 5.96-, and 6.74-fold increases in leaf B levels in Trifoliate orange, Ziyangxiangcheng, Carrizo citrange, and Red tangerine seedlings relative to respective controls. Similarly, stem B content in these seedlings rose by 5.70-, 2.20-, 2.44-, and 1.97-fold, respectively, while root B content rose by 2.94-, 2.58-, 3.52-, and 3.42-fold,

respectively. B content in Trifoliate orange leaves increased less substantially than in the three other analyzed rootstocks.

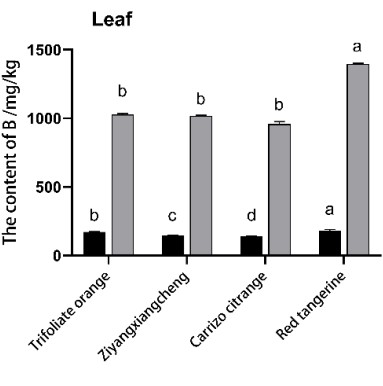
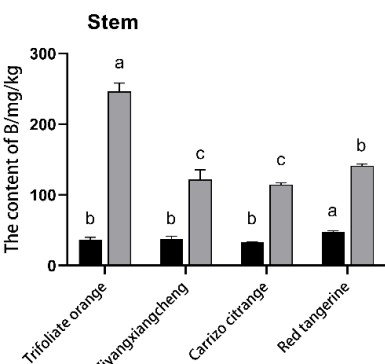
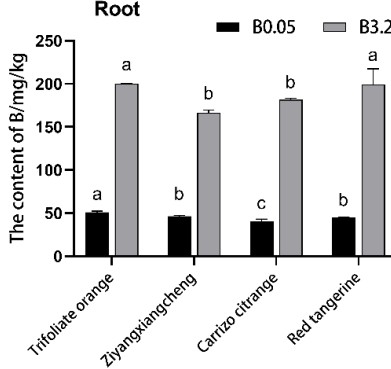

**Figure 4.** The impact of B 0.05 and B 3.2 conditions on B content in different four citrus rootstock seedlings. Note: B 0.05 served as the control (CK) concentration, and data were compared with Duncan's multiple range test. Different letters indicate significant differences between rootstocks at the same B concentration ($p < 0.05$).

### 3.7. Comprehensive Evaluation of Citrus Rootstock B Tolerance under Excess B Stress Conditions

Next, the B tolerance of these rootstocks was analyzed in a comprehensive manner based on seven indices (relative conductivity, Pn, SOD activity, POD activity, CAT activity, proline content, and soluble protein content) through a principal coordinate analysis approach. This result revealed that the cumulative contribution rate of the first two principal components (F1 and F2) was 93.05%, indicating that they were able to reflect the majority of the data included in these seven indicators. Relative conductivity, proline content, and SOD, POD, and CAT activity contributed to F1, while Pn and soluble protein content contributed to F2. The comprehensive value D corresponding to the B tolerance of these four rootstocks was also calculated (Table 7), revealing that the highest B tolerance was evident in Red tangerine (0.998), followed by Carrizo citrange (0.465), Ziyangxiangcheng (0.287), and Trifoliate orange (0.246) rootstocks.

**Table 7.** Comprehensive index (F), subordinate function (U), and integrated assessment (D) values for four citrus rootstock seedlings under excess B stress conditions.

| Rootstocks | Comprehensive Index | | Subordinate Function Value | | D |
|---|---|---|---|---|---|
| | F1 | F2 | U1 | U2 | |
| Trifoliate orange | −2.74 | 1.18 | 0.000 | 1.000 | 0.246 |
| Ziyangxiangcheng | −0.83 | −1.02 | 0.341 | 0.120 | 0.287 |
| Carrizo citrange | 0.71 | −1.32 | 0.616 | 0.000 | 0.465 |
| Red Tangerine | 2.86 | 1.16 | 1.000 | 0.992 | 0.998 |
| Index weight | | | 0.754 | 0.246 | |

## 4. Discussion

B is an essential micronutrient that can facilitate the growth and development of plants [28], but excessively high B levels can have toxic effects on plant physiology, adversely impacting photosystem II, carbohydrate metabolism, carbon dioxide assimilation, and antioxidant activity, thus, interfering with growth and dry-matter accumulation [4,29]. Under excess B stress conditions, citrus rootstock leaf tips turned yellow and leaves exhibited macular chlorosis along the leaf margin [30], eventually turning yellow as branches died, ultimately leading to plant death [31,32]. In this study, the stress of excess B reduced rootstock growth, CAT activity, soluble protein content, chlorophyll content, and photosynthetic capacity while increasing proline levels, SOD activity, POD activity, and relative conductivity. Of the analyzed rootstocks, Red tangerine exhibited the smallest reductions in biomass, SPAD, and Pn relative to controls under high B stress. These results indicated that excess B may adversely impact possible variation in Rubisco carboxylation and PSII efficiency, in line with results in pomelos and 'Xuegan' mandarins reported previously by indices et al. [33] and Huang et al. [34]. The degree of leaf yellowing and leaf photosynthetic indices are important indicators that can be used to gauge the severity of B toxicity. As such, comprehensive analyses of injury-related symptoms, B injury indices, and clinical parameters together revealed that Trifoliate orange was the most B-sensitive rootstock tested in this study, whereas Red tangerine exhibited the highest levels of adaptability under conditions of excessively high B stress.

The mechanisms by which plants respond to excessively high mineral element levels, particularly microelements, include reducing root-mediated mineral absorption, activating the ability of B to transport upwards in plants that metabolize these toxins [29,35]. Under conditions of excessively high B stress, the B levels in these four rootstocks rose, with the highest levels being evident in the leaves followed by the roots and stems, consistent with excess B stress impacting all of these tissues. While Red tangerine leaves and roots contained higher B levels than those of other rootstocks, Red tangerine growth, nonetheless, remains stronger under high B stress conditions. Similar results were obtained from previous studies that the leaves of 'Newhall' navel oranges grafted on Carrizo citrange, which accumulated more B than did the leaves of 'Bonanza' navel oranges, whereas 'Newhall' navel oranges exhibited higher levels of B tolerance [36]. The electrical conductivity of the leaves of each rootstock significantly increased with rising B concentrations, with the highest increase being evident in Trifoliate oranges. Red tangerine seedlings exhibited the lowest increase rate in electrical conductivity among the four rootstocks, indicating stronger membrane stability. In addition, rootstock growth was not affected by the levels of B in roots and leaves. These results suggest that the mechanisms governing excess B tolerance in plants differ from those for other common microelements, such as potassium and nitrogen, and may be related to the mechanisms governing membrane stability.

Excess B stress also induces reactive oxygen species production in plants, contributing to membrane lipid peroxidation and the consequent selective permeabilization of the cell membrane [37,38]. Antioxidative enzymes can scavenge these reactive species and, thereby, mitigate oxidative-stress-related damage [29]. Here, rising B concentrations were associated with significant increases in SOD and POD activity levels, as compared to controls in the leaves of all four citrus rootstocks. SOD and CAT are important antioxidant enzymes that can effectively reduce the damage caused by excessively high element levels in crops. Under high B stress conditions, the activity of SOD and CAT was the highest in Red tangerine and the lowest in Trifoliate orange, consistent with the results of Zhang et al. [3]. Increased SOD activity levels in Trifoliate orange seedlings were significantly less than those in Red tangerine, while reductions in CAT activity were significantly greater than in Red tangerine, thus, demonstrating the more robust antioxidant activity of Red tangerine as compared to Trifoliate orange. B-tolerant chickpea leaves have been reported to contain high levels of soluble protein and osmotic regulators, including proline, thereby contributing to better membrane stability and fluidity [39,40]. Under stress conditions, plants often accumulate high levels of soluble protein and proline so that they can maintain normal

osmotic pressure and reduce stress [41–43]. High B-induced stress levels consistently increased the soluble protein and proline content in all four rootstocks, with these changes being most pronounced in Trifoliate orange. However, higher B levels (B 3.2) exceed the tolerance of these plants, thereby inhibiting protein synthesis or increasing protein degradation. As such, Red tangerine that has stronger antioxidant activity and osmotic regulation ability resulted in the highest levels of B tolerance, as indicated by the stronger photosynthetic capacity and less toxic symptoms.

**5. Conclusions**

In summary, of the four tested citrus rootstocks, Red tangerine exhibited the highest levels of B tolerance, followed by Carrizo citrange and Ziyangxiangcheng, with Trifoliate orange exhibiting the poorest tolerance. Mechanisms governing membrane stability, rather than the transport and assimilation of B, were found to contribute to the tolerance of these citrus rootstocks for excess B stress conditions. As such, rootstock selection should be guided based on local soil conditions, and B application should be limited based on the type of rootstock in use to protect citrus orchards from damage resulting from excessively high soil B concentrations.

**Author Contributions:** Conceptualization, W.Y. and H.Y.; methodology, H.Y.; validation, W.Y. and H.Y.; formal analysis, W.Y. and H.Y.; investigation, L.L.; resources, L.P. and L.L.; writing—original draft preparation, W.Y. and H.Y.; writing—review and editing, L.P.; supervision, L.P.; project administration, L.P. and C.C.; funding acquisition, L.P. All authors have read and agreed to the published version of the manuscript.

**Funding:** This research was funded by the National Key Research and Development Program of China "Integration and demonstration of high-quality, light, simple and efficient cultivation techniques for evergreen fruit trees" (2020YFD100), Ministry of Finance and Ministry of Agriculture and Rural Affairs: National Modern Agricultural Industry Technology System.

**Institutional Review Board Statement:** Not applicable.

**Informed Consent Statement:** Not applicable.

**Data Availability Statement:** All data are contained within the article.

**Conflicts of Interest:** The authors declare no conflict of interest.

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
