# Peer review of "Tolerance and Physiological Responses of Citrus Rootstock Cultivars to Boron Toxicity"

_horticulturae, doi:10.3390/horticulturae9010044_

Round 1

Reviewer 1 Report

Regression analysis is the valid statistical analysis tool to use for quantitative variables, not analysis of variance and Duncan's multiple range test.

Why the B concentrations were not incremental?

Author Response

Thank you for your comments. We have responded to your specific comments and suggestions using a blue font as detailed below.

Regression analysis is the valid statistical analysis tool to use for quantitative variables, not analysis of variance and Duncan's multiple range test.

Thank you for your suggestion. While we tried to perform regression analyses for all variables, these results were not ideal due to the different samples and treatment times. As such, we instead sought to reveal the differences at the levels of rootstocks and treatments using ANOVAs and Duncan's multiple range test. In our subsequent experimental design will consider how best to incorporate regression analyses.

Why the B concentrations were not incremental?

Thank you for raising this concern. According to the studies of Aparicio-Duran et al., Wang et al. and Sheng et al., we performed B treatment and standard concentrations. These reports indicated that the appropriate concentration of B is 0.05 mM for growth, while B concentrations beyond 2.5mM can cause serious toxicity. We thus use 0.05 mM as the control concentration, with a 4-fold increase, and a maximum concentration of 3.2 mM.

Reference:

Aparicio-Durán L, Frederick G. Gmitter J, Arjona-López J M, Grosser J W, Elázquez R C-V, Hervalejo Á, Arenas-Arenas F J. Evaluation of three new citrus rootstocks under boron toxicity conditions [J]. Agronomy, 2021, 11: 2049-2062. DOI: 10.3390/agronomy11122490

Wang N N, Peng S A, Liu Y Z. Advances on boron nutrition of citrus [J]. Journal of Huazhong Agricultural University, 2015, 34(4): 137-143. DOI: 10.13300/j.cnki.hnlkxb.2015.04.023

Sheng O, Song S W, Peng S A, Deng X X, Lu X P. Effects of different Boron concentrations on growth, boron absorption and distribution of Navel orange [C] China Citrus Science and Technology Innovation and Industrial Development Strategy Forum and China Citrus Society Annual Meeting 2007 Proceedings.

Reviewer 2 Report

The manuscript “Tolerance and physiological responses of citrus rootstock cultivars to boron toxicity” explore the tolerance and physiological responses of four citrus rootstocks to excess B levels, to contribute to the identification of key determinants of B tolerance and, consequently, to support rootstock selection efforts for citrus production.  So, the title reflects properly the research presented. 

The abstractintroduction, and Material and Methods give a concise but appropriate context of the study purpose. Only few comments could be found in attachement.

Concerning Material and Methods I have several questions about seedlings growth condition, number of seedlings, drought stress and rehydration, etc, specifically:

Results are interesting and adequately presented. However, I think tables 1-2 would be improved if the different levels of Boron concentrations were separated by lines, and tables 3-7 separated by rootstocks.

In the Discussion, the authors affirm “Excess B was also able to destroy cellular chloroplasts and impact carboxylation reaction efficiency, adversely impacting photosystem II photosynthetic efficiency and chlorophyll content, in line with results in pomelos and ‘Xuegan’ mandarins reported previously by indices et al. [33] and Huang et al.”. These are not results presented by the authors. So, I suggest that instead of an affirmation the authors could discuss the Pn alteration in term of possible variation in Rubisco carboxylation and PSII efficiency. Also, discussion about antioxidant enzymes activity in the four rootstocks should be improved.

One of the main conclusions is that “Mechanisms governing membrane stability, rather than the transport and assimilation of B, were found to contribute to the tolerance of these citrus rootstocks for excess B stress conditions”. This conclusion must be discussed and supported with the results of relative leave conductivity and B level in different part of the plants, not done by the authors as far as I notice.

See, please, the revised manuscript in attachment

Author Response

Thank you for your comments. We have responded to your specific comments and suggestions using a blue font as detailed below.

The manuscript “Tolerance and physiological responses of citrus rootstock cultivars to boron toxicity” explore the tolerance and physiological responses of four citrus rootstocks to excess B levels, to contribute to the identification of key determinants of B tolerance and, consequently, to support rootstock selection efforts for citrus production. So, the title reflects properly the research presented.

The abstract, introduction, and Material and Methods give a concise but appropriate context of the study purpose. Only few comments could be found in attachment. Concerning Material and Methods I have several questions about seedlings growth condition, number of seedlings, drought stress and rehydration, etc., specifically:

Thank you for your suggestion. In the Materials and Methods section 2.1, we have added additional details as follows regarding B: “The B solution used for treatment was analytical pure borax (Na2B4O7·10H2O, Sinopharm Chemical Reagent Co. LTD)”, number of seedlings: “sixty neat rootstock seedlings of similar size were selected for each rootstock”, seed source: “The ripe fruits of four rootstocks were collected from the National Citrus Germplasm Resources Nursery from Southwest University. The seeds were collected, the pectin was removed with quicklime, and they were then washed for later use”, and light conditions: “photon photosynthetic flux density (PPFD) of 150 μmol.s-1.m-2”.

Results are interesting and adequately presented. However, I think tables 1-2 would be improved if the different levels of Boron concentrations were separated by lines, and tables 3-7 separated by rootstocks.

We appreciate your comments. As per your suggestions, we have separated the different levels of rootstocks and B concentrations in Tables 1-2 and 3-7 using lines.

In the Discussion, the authors affirm “Excess B was also able to destroy cellular chloroplasts and impact carboxylation reaction efficiency, impacting photosystem II photosynthetic efficiency and chlorophyll content, in line with results in pomelos and ‘Xuegan’ mandarins reported previously by indices et al. [33] and Huang et al.”. These are not results presented by the authors. So, I suggest that instead of an affirmation the authors could discuss the Pn alteration in term of possible variation in Rubisco carboxylation and PSII efficiency.

Thank for your excellent suggestion. We have discussed the contents in the revised manuscript as follows: “Of the analyzed rootstocks, Red tangerine exhibited the least reductions in biomass, SPAD, and Pn relative to controls under high B stress. These results indicated that excess B may adversely impact possible variation in Rubisco carboxylation and PSII efficiency, in line with results in pomelos and ‘Xuegan’ mandarins reported previously by Indices et al. and Huang et al.” in the Discussion section.

Reference:

Gimeno, Simon, Nieves, Martinez, Camara-Zapata J M, Garcia A L, Garcia-Sanchez. The physiological and nutritional responses to an excess of boron by Verna lemon trees that were grafted on four contrasting rootstocks [J]. Trees-Structure and Function, 2012, 26(5): 1513-1526. DOI: 10.1007/s00468-012-0724-5

Huang J H, Cai Z J, Wen S X, Guo P, Ye X, Lin G Z, Chen L S. Effects of boron toxicity on root and leaf anatomy in two Citrus species differing in boron tolerance [J]. Trees-Structure and Function, 2014, 28(6): 1653-1666. DOI: 10.1007/s00468-014-1075-1

Also, discussion about antioxidant enzymes activity in the four rootstocks should be improved.

Thank you for your suggestion. We have clarified the role of antioxidant activity in the tolerance of excess B in the Discussion section as follows:

“SOD and CAT are important antioxidant enzymes that can effectively reduce the damage caused by excessively high element levels in crops. Under high B stress conditions, the activity of SOD and CAT was the highest in Red tangerine and the lowest in Trifoliate orange, consistent with the results of Zhang et al.”

Reference:

Nable R O, Bañuelos G, Paull J G. Boron toxicity [J]. Plant & Soil, 1997: 181-198.

Zhang J, Lin W J, Li X B, Zhan Y Y, Zhang S C, Li Y. Symposium of boron toxicity and photosynthesis response study in leaves of Ponkan (Citrus reticulata Blanco) [J]. Journal of Plant Nutrition and Fertilizers, 2020, 26(10): 1879-1886. DOI: 10.11674/zwyf.20123

One of the main conclusions is that “Mechanisms governing membrane stability, rather than the transport and assimilation of B, were found to contribute to the tolerance of these citrus rootstocks for excess B stress conditions”. This conclusion must be discussed and supported with the results of relative leave conductivity and B level in different part of the plants, not done by the authors as far as I notice.

Thank you for your suggestion. We have already added some evidence of relative leaf conductivity and B level in different parts of these plants to support our Discussion, as detailed below. The contents in parentheses are in the original manuscript.

(While Red tangerine leaves and roots contained higher B levels than those of other rootstocks, Red tangerine growth nonetheless remains stronger under high B stress conditions. Similar results were obtained from previous studies that the leaves of ‘Newhall’ navel oranges grafted on Carrizo citrange, which accumulated more B than did the leaves of ‘Bonanza’ navel oranges, whereas ‘Newhall’ navel oranges exhibited higher levels of B tolerance.) “The electrical conductivity of the leaves of each rootstock significantly increased with rising B concentrations, with the highest increase being evident in Trifoliate oranges. Red tangerine seedlings exhibited the lowest increase rate in electrical conductivity among the four rootstocks, indicating stronger membrane stability. In addition, rootstock growth was not affected by the levels of B in roots and leaves. These results suggest that the mechanisms governing excess B tolerance in plants differ from those for other common microelements such as potassium and nitrogen, and may be related to the mechanisms governing membrane stability.”  

Reference:

Wang N N, Peng S A, Liu Y Z. Advances on boron nutrition of citrus [J]. Journal of Huazhong Agricultural University, 2015, 34(4): 137-143. DOI: 10.13300/j.cnki.hnlkxb.2015.04.023

See, please, the revised manuscript in attachment

Thank you for the detailed reviews. In the Material and Methods sections 2.1 and 2.3, we have changed the “a light intensity of 10000LX” to “photon photosynthetic flux density (PPFD) of 150 μmol.s-1.m-2”, and also modified the method used to measure soluble protein levels to the Bradford method.

Some details have been incorporated into the revised manuscript as per your suggestions.

Reviewer 3 Report

The submitted manuscript deals with the issue of the effect of boron toxicity on citrus rootstocks. From the grower's point of view, this is probably an important current, but only local problem, with regard to the citrus growing area in the world. The introduction is sufficient, including the stated objectives of the work. The methodological part needs to be modified, because I lack information about which form of boron it was. What was the source and origin of the seed. I believe that the information that Excel was used for the charts is completely irrelevant. Furthermore, I lack information about which statistical methods were used. Due to the fact that correlation relations were created, less attention is paid to them in the results section itself. As part of the results, I recommend changing the size of the graphs and the color of the curves, as it is sometimes somewhat confusing. At the same time, correlations or other relationships could be established between individual monitored parameters. As part of the description of the visual symptoms of toxicity B, it would be appropriate to supplement this with, for example, photographs. The discussion is rather general and descriptive. Authors cite used sources inconsistently – lowercase letters, journal name abbreviations, full title, etc. Are all cited sources required to be marked [D]?

Author Response

Thank you for your comments. We have responded to your specific comments and suggestions using a blue font as detailed below.

The submitted manuscript deals with the issue of the effect of boron toxicity on citrus rootstocks. From the grower's point of view, this is probably an important current, but only local problem, with regard to the citrus growing area in the world. The introduction is sufficient, including the stated objectives of the work.

The methodological part needs to be modified, because I lack information about which form of boron it was. What was the source and origin of the seed.

Thank you for this input. In the Methods, the boron solution was analytical pure borax (Na2B4O7·10H2O, Sinopharm Chemical Reagent Co. LTD), and the seeds were harvested at the National Citrus Germplasm Resources Nursery from Southwest University. In addition, we have added some information regarding the culture conditions of our seedlings, such as light conditions.

I believe that the information that Excel was used for the charts is completely irrelevant.

Thank you for your suggestion. We have modified the descriptions in our Materials and Methods section 2.5 as follows:

“Data analyses and principal component analyses were performed using SPSS 19.0. Results were compared with ANOVAs and Duncan’s multiple range test. GraphPad Prism 9.2 was used for figure construction.”

Furthermore, I lack information about which statistical methods were used. Due to the fact that correlation relations were created, less attention is paid to them in the results section itself.

Thank you for your suggestion. We have added detailed statistical methods to Materials and Methods section 2.5. “Results were compared with ANOVAs and Duncan’s multiple range test.” As this analysis yielded some new results, additional descriptions of significance have been provided in the Results.

As part of the results, I recommend changing the size of the graphs and the color of the curves, as it is sometimes somewhat confusing. At the same time, correlations or other relationships could be established between individual monitored parameters.

Thank you for your suggestion. We have changed the color and size of the chart. Data in figure 2 were compared with Duncan’s multiple range test. Different letters indicate significant differences between rootstocks at the same DAT (P < 0.05).

As part of the description of the visual symptoms of toxicity B, it would be appropriate to supplement this with, for example, photographs.

Thank you for your suggestion. We have added the  figure in section 3.1 of the Results (Figure 1). and have renumbered subsequent figures accordingly.

The discussion is rather general and descriptive.

Thank you for your critique. We have added some details information regarding enzyme activity, membrane stability, B levels in different parts of the plants, and photosynthesis in the Discussion section of the revised manuscript.

“Of the analyzed rootstocks, Red tangerine exhibited the least reductions in biomass, SPAD, and Pn relative to controls under high B stress. These results indicated that excess B may adversely impact possible variation in Rubisco carboxylation and PSII efficiency, in line with results in pomelos and ‘Xuegan’ mandarins reported previously by Indices et al. and Huang et al.”

“The electrical conductivity of the leaves of each rootstock significantly increased with rising B concentrations, with the highest increase being evident in Trifoliate oranges. Red tangerine seedlings exhibited the lowest increase rate in electrical conductivity among the four rootstocks, indicating stronger membrane stability. In addition, rootstock growth was not affected by the levels of B in roots and leaves. These results suggest that the mechanisms governing excess B tolerance in plants differ from those for other common microelements such as potassium and nitrogen, and may be related to the mechanisms governing membrane stability.”

“SOD and CAT are important antioxidant enzymes that can effectively reduce the damage caused by excessively high element levels in crops. Under high B stress conditions, the activity of SOD and CAT was the highest in Red tangerine and the lowest in Trifoliate orange, consistent with the results of Zhang et al.”

Reference:

Gimeno, Simon, Nieves, Martinez, Camara-Zapata J M, Garcia A L, GARCIA-SANCHEZ. The physiological and nutritional responses to an excess of boron by Verna lemon trees that were grafted on four contrasting rootstocks [J]. Trees-Structure and Function, 2012, 2012,26(5) (-): 1513-1526. DOI: 10.1007/s00468-012-0724-5

Huang J H, Cai Z J, Wen S X, Guo P, Ye X, Lin G Z, Chen L S. Effects of boron toxicity on root and leaf anatomy in two Citrus species differing in boron tolerance [J]. Trees-Structure and Function, 2014, 28(6): 1653-1666. DOI: 10.1007/s00468-014-1075-1

Wang N N, Peng S A, Liu Y Z. Advances on boron nutrition of citrus [J]. Journal of Huazhong Agricultural University, 2015, 34(4): 137-143. DOI: 10.13300/j.cnki.hnlkxb.2015.04.023

Zhang J, Lin W J, Li X B, Zhan Y Y, Zhang S C, Li Y. Symposium of boron toxicity and photosynthesis response study in leaves of Ponkan (Citrus reticulata Blanco) [J]. Journal of Plant Nutrition and Fertilizers, 2020, 26(10): 1879-1886. DOI: 10.11674/zwyf.20123

Authors cite used sources inconsistently – lowercase letters, journal name abbreviations, full title, etc. Are all cited sources required to be marked [D]?

Thanks for your suggestion. We have updated our references to address the errors that you noted. Please refer to the attached latest manuscript for details.

Round 2

Reviewer 2 Report

I have no comments or suggestions to this second version of the menuscript